# A Simulation Study on the Smoke Control Effect with Different Smoke Exhaust Patterns and Longitudinal Air Supply for Ultra-Wide Tunnels

**Ying Li [1], Fang Huang [2], Chuyuan Ma [3] and Kaixuan Tang [3],\***

[1] College of Power Engineering, Naval University of Engineering, Wuhan 430074, China; zero789@sohu.com
[2] Hubei Huanggang Polytechnic of Emergency Management, Huanggang 438000, China; huangfang_0504@163.com
[3] School of Safety Science and Emergency Management, Wuhan University of Technology, Wuhan 430070, China; docmcy@whut.edu.cn
\* Correspondence: kaixuant@whut.edu.cn

**Abstract:** This study was motivated by the lack of understanding of the smoke control effect on an ultra-wide tunnel fire, with different smoke exhaust patterns (sidewall and top exhaust patterns) and longitudinal air supply volume (0, 30%, 50%, 70%, and 90%). A full-scale ultra-wide tunnel model was constructed based on the FDS and the fire parameters were analyzed, such as the longitudinal spread distance of smoke, the smoke layer height and the temperature at safe height. In addition, the smoke exhaust efficiency was calculated based on the mass flux of $CO_2$, and the smoke control effect with different smoke exhaust patterns and air supply volumes was compared. Results show that the smoke exhaust patterns and air supply ratios have a great influence on smoke spread distance and exhaust efficiency. The smoke spread distance is shortened by increasing the longitudinal air supply volume, and when the ratio of air supply volume to smoke exhaust volume is less than 50%, the top exhaust pattern can control the spread of smoke better with a smaller smoke spread distance. In addition, the height of the smoke layer is controlled above the safe height of 2 m under the top smoke exhaust, and the temperature at both ends of the tunnel (25 °C) is lower than that under the sidewall exhaust pattern (35 °C). The smoke exhaust efficiency was calculated based on the mass flow rate of $CO_2$, and the exhaust efficiency of the top exhaust pattern (~70%) is significantly higher than that of the sidewall exhaust pattern (~55%). However, as the air supply volume increases, there is a reduced increase in the exhaust efficiency. Therefore, taking the economic cost into account, the air supply ratios of 30% and 50% are the best for top and sidewall exhaust patterns, respectively. The results of this work provide important information about smoke distribution characteristics in an ultra-wide tunnel fire and may guide its design of smoke exhaust.

**Keywords:** ultra-wide tunnel fire; smoke control; top exhaust pattern; longitudinal air supply; exhaust efficiency

## 1. Introduction

In order to solve the transportation connection between bays and land, the application of immersed tunnels is becoming more and more extensive, and it is developing towards extra-long and ultra-wide. However, there is a high fire risk in the underwater tunnel because of its long and narrow closed structure and the complex traffic environment. The main dangers of fire in an underwater tunnel are high temperatures and toxic smoke, which can lead to injury and death [1]. Therefore, smoke control is the key to decreasing the fire hazard of tunnels, and it has attracted much attention [2–4].

Over the past few decades, studies on the design of smoke exhaust in tunnel fires have been extensively reported [5–10]. It is an effective method to explore the effects of different smoke designs through field experiments in tunnels, but it has high complexity

and economic cost. Thus, most of the research was performed with numerical simulation. For instance, the effects of the ventilation condition [11,12] and the layout of exhaust vents (including the location, size and number of vents) [13,14] on the temperature, smoke distribution and visibility were extensively studied. A series of papers [15–18] analyzed the issue of the efficiency of ventilation through simulation and experiments. Oucherfi et al. [16] evaluated the efficiency of a transversal ventilation system by simulation based on the buoyancy fluxes and concluded that ventilation rate is the most influential parameter in efficiency. In their experiments, Chaabat et al. [18] studied the confinement of smoke flow between adjacent exhaust vents and quantified the influence of the shape and the position of the dampers. Mechanical ventilation has widely been used in tunnel fires, which is an important part of smoke control design, and the layout of vents (including exhaust vents and air supply vents) has a significant impact on the smoke exhaust effect, which can be characterized by smoke exhaust efficiency.

However, the effectiveness and applicability of the smoke exhaust design are rarely studied from the perspective of smoke exhaust efficiency. Xu et al. [19] carried out many fire tests in tunnels to explore the smoke exhaust efficiency in longitudinal and transverse smoke exhaust systems, and it was found that the exhaust efficiency reaches 35% and 50%, respectively, for transverse longitudinal smoke exhaust. In addition, the exhaust efficiency is improved when the number and area of the exhaust vents in the tunnel are increased. Zhu et al. [20] used CFD simulation software to carry out a fundamental study on the influence of the layout of exhaust vents on the exhaust efficiency, considering the distance between adjacent exhaust vents, and the location and size of vents. It was found that the location and area of the exhaust vents are major factors in determining the exhaust efficiency, and the heat and smoke exhaust efficiency are dependent on the layout and distance of the vents. Furthermore, the smoke exhaust patterns also have an important influence on the smoke efficiency. Liu et al. [21] analyzed the smoke efficiency of exhaust patterns with different opening modes of vents, which is based on the results of many full-scale fire experiments in a three-lane underwater tunnel. The higher ventilation rate and number of vents significantly improved the exhaust efficiency. However, these studies were performed using conventional tunnels (such as shield tunnels), and few of them involved ultra-wide tunnels. In fact, smoke can spread more widely in ultra-wide tunnels, posing new challenges in the smoke control of tunnel fires. Buchanan et al. [22] and Chen et al. [23] found that it is difficult to achieve the expected effect of smoke exhaust for an ultra-wide tunnel through the traditional smoke exhaust design of setting different layouts of vents, and proposed a coupling method of mechanical smoke exhaust and air supply. In addition, the opening direction of vents, up or sideways (corresponding to top and sidewall exhaust patterns), has different smoke exhaust effect, but it is rarely explored.

To fill this gap, this paper studied the smoke control effect for an ultra-wide tunnel fire based on FDS simulation. The effects of smoke exhaust patterns and longitudinal air supply volume on the smoke spread distance, smoke layer height, temperature at the safe height, and smoke exhaust efficiency in the tunnel were analyzed based on the numerical results.

## 2. Numerical Simulation

### 2.1. Model Tunnel

The Fire Dynamics Simulator (FDS 6.7) is extensively used in the simulation calculation of tunnel fires to explore the smoke control effect of its smoke spread and temperature distribution, and its applicability and accuracy have been verified [24–26]. The Large Eddy Simulation (LES) method was used in this study, including conservations of mass, momentum, energy and species, with the combustion, turbulence and radiation models set as default [27,28]. The main governing equations are as follows:

Conservation of mass:

$$\frac{\partial \rho}{\partial t} + \nabla \cdot (\rho \mathbf{u}) = 0 \tag{1}$$

Conservation of momentum:

$$\rho\left(\frac{\partial \mathbf{u}}{\partial t} + (\mathbf{u}\cdot\nabla)\mathbf{u}\right) + \nabla p - \rho g = \mathbf{f} + \nabla\cdot\tau \tag{2}$$

Conservation of energy:

$$\frac{\partial}{\partial t}(\rho h) + \nabla\cdot\rho h\mathbf{u} = \frac{\partial p}{\partial t} + \mathbf{u}\cdot\nabla \mathbf{p} - \nabla\cdot q_r - \nabla(k\nabla T) + \sum_i \nabla(h_i \rho D_i \nabla Y_i) \tag{3}$$

Conservation of species:

$$\frac{\partial}{\partial t}(\rho Y_i) + \nabla\cdot\rho Y_i\mathbf{u} = \nabla\cdot\rho D_i \nabla Y_i + \dot{m}_i'' \tag{4}$$

where $\rho$ is the gas density, kg/m$^3$; $t$ is the time, s; $u$ is the velocity vector, m/s; $p$ is the pressure, Pa; $g$ is the gravitational acceleration, m/s$^2$; $f$ is the external force vector, N; $\tau$ is the stress tensor, N; $h$ is the sensible enthalpy, J/kg; $q_r$ is the heat release rate per unit volume from a chemical reaction, W/m$^2$; $D_i$, $Y_i$, and $\dot{m}_i''$ are the diffusion coefficient, mass fraction and loss rate of unit volume.

Figure 1 shows the physical model of the tunnel with different smoke exhaust patterns. In this paper, the tunnel model was constructed based on an ultra-wide cross-sea tunnel, which is twin-bore with eight traffic lanes. To reduce the computing cost, a single-bore tunnel model was constructed, as shown in Figure 1, with dimensions of 600 m in length, 18 m in width and 7.5 m in height, respectively. The top, bottom, and sidewalls of the tunnel were set as "INERT". The entrance and exist of the tunnel were parallel to the XOZ plane. A "VENT" surface was arranged at the cross-section of the entrance (y = 0 m) and exit (y = −600 m), which the directions of the airflow were positive and negative along the Y-axis. Both ends of the tunnel were opened to the environment, simulating the open space outside the tunnel. The ambient temperature was set to 293 K. In addition, the simulation calculation time was set to 1200 s to ensure that the smoke could adequately spread and the calculation amount is reduced to improve the accuracy.

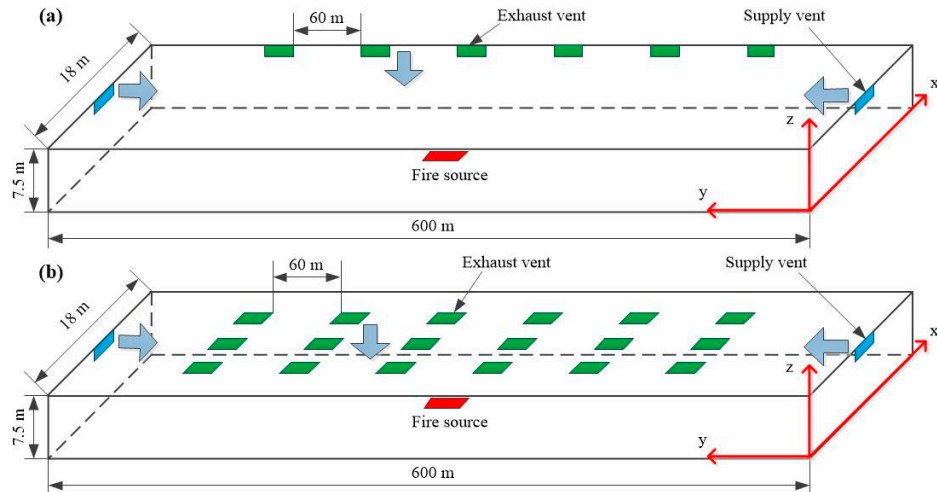

**Figure 1.** Schematic diagram of tunnel model with different exhaust patterns: (**a**) sidewall; (**b**) top.

The fire source was set in the center of the single-bore tunnel (x = 300 m, y = 9 m, z = 0 m). The length and width of the fire source were 5 m and 2 m, respectively, with a surface of 10 m$^2$. The tunnel fire was simulated by the burning of diesel, and the chemical formula for fuel was simplified as C$_{12}$H$_{23}$ (C is carbon and H is hydrogen). Previous

research [29] reported that the power of a tunnel fire can reach its maximum value within a very short time. It is a relatively quick $t^2$ combustion model, which can be expressed as:

$$Q = \alpha t^2 \tag{5}$$

where $Q$ is the fire power, kW; $\alpha$ is the fire increasing modulus with a value of 0.1876 [24], and $t$ represents the duration time of fire growing, s. In this paper, the fire power was set to 20 MW, which is equivalent to the heat release rate of a coach fire. Therefore, the fire scenario was constructed on the basis of a coach fire occurring in the tunnel.

For the sidewall smoke exhaust pattern (as Figure 1a), the exhaust vents, with dimensions of 4 m in length and 1.5 m in height, were arranged in the sidewall of the tunnel (x = 18 m) and the number of vents was 6. The distance from the bottom of the exhaust vent to the carriageway was 6 m (z = 6 m). For the top smoke exhaust pattern (as Figure 1b), 3 rows of exhaust vents were arranged at the top of the tunnel (z = 7.5 m). Each row has 6 smoke vents. The length and width of the exhaust vent are 4 m and 1.5 m, respectively. In addition, these smoke vents were symmetrically distributed with the central surface of the fire source, and they were 138 m from the ends of the tunnel. In addition, the tunnel model is devoid of vehicles, ignoring the obstacles to simplify the simulation calculation.

The difference between the two exhaust patterns is not only reflected in the layout of the vents, including the location and number of vents, but also in the exhaust volume rate. The determination of the exhaust volume rate was based on the plume model proposed by Heskestad [30], and the total exhaust volume was set at 120 $m^3$/s for both patterns. The velocity boundary conditions at the exhaust vents were set at 3.33 m/s for the sidewall pattern and 1.11 m/s for the top pattern, respectively. At the same time, five ratios of air supply volume rate to exhaust volume rate, 0, 30%, 50%, 70% and 90%, were selected to study the influence of this ratio on smoke control in tunnel fires. Therefore, the boundary conditions of the air-supply vents with different ratios were set as 0, 3, 5, 7, and 9 m/s respectively.

In addition, many measuring points were arranged to monitor the smoke temperature, the smoke layer height and the mass flux of $CO_2$, as follows:

(1) Layer zoning device: 61 Layer zoning devices were set in the longitudinal centerline of the tunnel with an interval at 10 m to monitor the smoke layer height.

(2) Thermocouples: To monitor the temperature variation at a safe height (2 m), a series of thermocouples were installed every 10 m along the longitudinal centerline of the tunnel, 2 m above the ground. In addition, near the fire source within 10 m, the thermocouples were arranged at a longitudinal interval of 1 m.

(3) $CO_2$: The mass flux of $CO_2$ around each exhaust vent was monitored using the parameter "MASS FLUX Z", SPEC-ID = "carbon dioxide". Coordinate parameters were equal to those of the exhaust vents.

### 2.2. Mesh Size

Mesh size is the most important aspect for numerical simulation because it determines the reliability and accuracy of the results. In general, a smaller mesh size results in more precise computation results, but it will increase the computing amount and economic cost. For FDS simulation, when the mesh is less than 0.1 of the characteristic diameter $D^*$, it can make sure that the results are acceptable and reliable [31]. The characteristic diameter can be determined by [32]:

$$D^* = \left[ \frac{Q}{\rho_\infty c_p T_\infty \sqrt{g}} \right]^{2/5} \tag{6}$$

where $Q$ is the heat release rate of fire, kW; $\rho_\infty$ is the density of ambient air, kg/$m^3$; $c_p$ is the specific heat capacity of air at constant pressure, J/kg/K); $T_\infty$ is the temperature of ambient air, K; and $g$ is the gravitational acceleration, m/$s^2$. $D^*$ is calculated to be 3.1 m when the heat release rate of fire is 20 MW, thus 0.1 $D^*$ is approximately 0.31 m, and then the calculated mesh size is smaller than smoke layer height and vent size (~2 m). The fire source is more sensitive to the density of mesh, thus the mesh size near the fire source (275–325 m)

is divided into $0.25 \times 0.25 \times 0.25$ m, and the mesh size in other area is $0.5 \times 0.25 \times 0.25$ m. Mesh distribution is shown in Table 1. The convergence criteria were set as $10^{-4}$.

**Table 1.** Distribution of mesh size.

| Fire Power | 0.1 *D*\* | Position | Mesh Size | Total Number of Mesh |
|---|---|---|---|---|
| 20 MW | 0.31 m | 0–275 m | $0.5 \times 0.5 \times 0.5$ m | 1,227,400 |
| | | 275–325 m | $0.25 \times 0.25 \times 0.25$ m | |
| | | 325–600 m | $0.5 \times 0.5 \times 0.5$ m | |

## 3. Results and Discussion

### 3.1. Smoke Spread

A large amount of high-temperature, toxic and harmful smoke is produced in the tunnel fire, which poses great threat to people. Therefore, it is important to understand the distribution of smoke and temperature for smoke exhaust design. Figure 2 shows the temperature distribution in the central surface under different air supply ratios. It is observed that the temperature distribution is symmetrical to the fire source and that the temperature at the top of the tunnel (200 °C) is much higher than that at the bottom of the tunnel. The difference in temperature is mainly attributed to the spread and distribution of smoke. In fact, the temperature distribution of the tunnel center surface can represent the smoke spread and its distribution in the tunnel. In a tunnel fire, the smoke first spreads upward because of the thermal buoyancy and then spreads around after encountering the tunnel ceiling. The smoke accumulates in the top of the tunnel and then descends downward, resulting in the difference between the upper and lower regions. In addition, the distance from the fire source to the temperature front of 60 °C, defined as the smoke spread distance, is continuously reduced when increasing the longitudinal air supply ratio. This is mainly caused by two aspects, on the one hand, the supplementary fan keeps supplying air into the tunnel and blowing the smoke closer to the smoke exhaust vent, which can exhaust smoke quickly. On the other hand, the pressure difference in the tunnel is decreased because of the increase in air supply ratio, which enhances the smoke exhaust efficiency.

For different smoke exhaust patterns, sidewall exhaust and top exhaust, the smoke spread distance shows some differences, as shown in Figure 3. When the air supply volume is less than 50% of the smoke exhaust volume, the smoke spread distance is about 10 m farther for the sidewall exhaust pattern than that of the top exhaust pattern, which means the sidewall exhaust pattern has a better smoke control effect under low air supply ratio (<50%). However, there is an opposite smoke control effect when the ratio of air supply is 70%. That is, the smoke spread distance for the top exhaust pattern is higher than that for the sidewall exhaust pattern. The smoke control effect is better for the top exhaust pattern with an air supply ratio of 70%. The spread of smoke can be controlled to a smaller area (−196~196 m), and the smoke spread distance for two exhaust patterns is similar after the air supply ratio is increased to 90%. In addition, when the air supply increases to 70% of the smoke exhaust, for sidewall pattern, the reduction of smoke spread distance is obvious (from 230 m to 210 m), but this obvious reduction is observed when it is increased to 90% for top pattern (from 218 m to 196 m). It can be seen that the smoke control effect has a significant improvement with a smaller air supply ratio for the sidewall exhaust pattern, but this significant improvement for the top exhaust pattern occurs when the air supply ratio is larger.

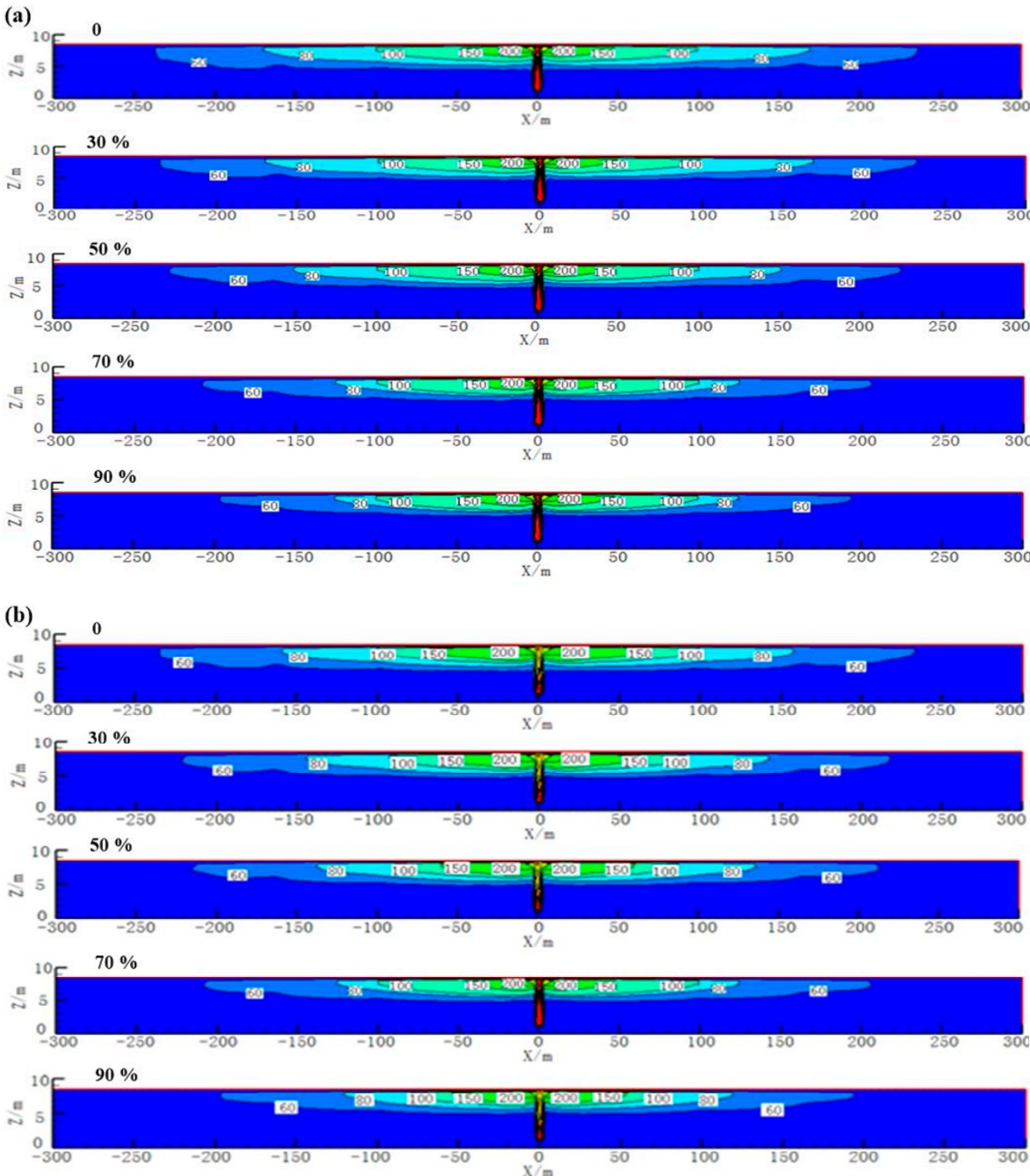

**Figure 2.** Smoke temperature distribution in central surface under different air supply volume: (**a**) sidewall exhaust; (**b**) top exhaust.

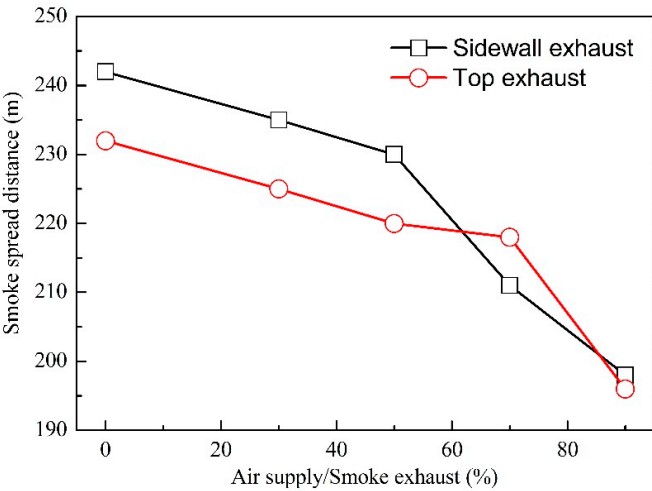

**Figure 3.** Variation of smoke spread distance with air supply for different smoke exhaust patterns.

### 3.2. Smoke Layer Height

In general, if the smoke cannot be extracted in time, it will accumulate in the top tunnel to form the smoke deposition phenomenon. The smaller smoke layer height is accompanied by a more serious smoke deposition and worsened smoke exhaust effect. Thus, the smoke layer height is an important parameter to characterize the smoke deposition. Figure 4 shows the distribution of the smoke layer height on the central surface of the tunnel, which presents as a "W" shape with the variation of the distance from the fire source. It is observed that for the two exhaust patterns, the distribution of smoke layer height is symmetrical to the fire source, and the smoke layer height near the fire source is significantly higher than both ends because of the thermal buoyancy. In addition, the height of the smoke layer near the fire source has little difference between the two exhaust patterns, and the distribution of the smoke layer height has a consistent trend with different longitudinal air supply. However, the effects of air supply ratios and smoke exhaust patterns on the smoke layer height at the ends of the tunnel can be distinguished.

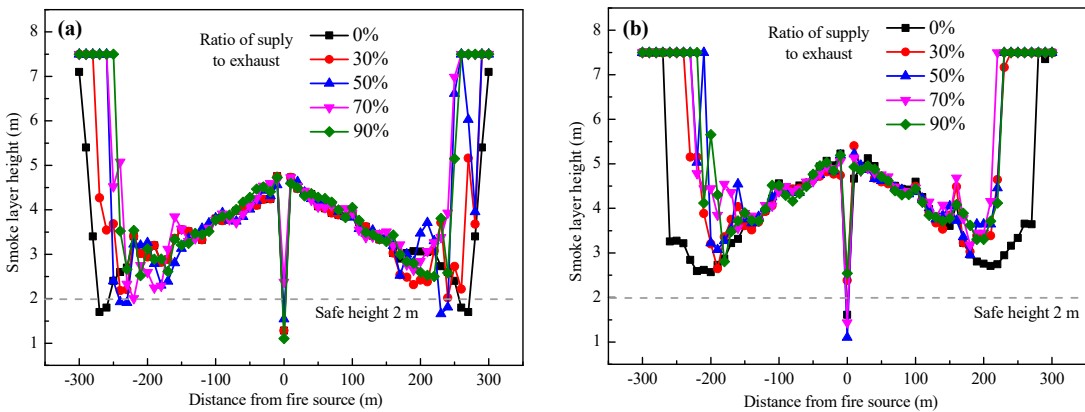

**Figure 4.** Smoke layer height under different air supply volume: (**a**) sidewall exhaust; (**b**) top exhaust.

For the sidewall exhaust pattern, the smoke layer height at the end of the tunnel (−300 m~−150 m and 150 m~300 m) is close to the safe height (2 m, the maximum height that the human eye can tolerate [33]), especially if it is less than the safe height with a lower air supply ratio (<50%). It means that a larger air supply ratio is more conductive to the smoke exhaust, so that the smoke deposition is reduced. However, the smoke layer height of the top exhaust pattern is obviously higher than the safe height, including in the absence of longitudinal air supply. When the longitudinal air supply ratio is small (30~50%), the smoke layer height at the ends of the tunnel is smaller, especially if there is no longitudinal

air supply, which is attributed to the decrease in pressure difference and the strong blowing effect. It can be seen that the smoke deposition is significantly weakened, and the smoke layer height is above the safe height (2 m) in the top exhaust pattern, but it occurs with a higher air supply ratio in the sidewall exhaust pattern, which means the top exhaust pattern has a better smoke control effect.

### 3.3. Temperature at Safe Height

The harm induced by high-temperature smoke may be caused by direct contact and heat radiance. The critical temperature of 68 °C was selected in this study as it is a harmful temperature to the human body, at which the human body will be irreparably damaged [33]. In general, the temperature inside the tunnel rises sharply because of the movement and accumulation of smoke. Figure 5 shows the temperature distribution at the safe height (2 m) with different air supply ratios and the comparison of the smoke exhaust effects between the two smoke exhaust patterns is also carried out in terms of personnel escape. It was found that the influence of air supply ratio on the temperature is not obvious, and the temperature curves under different longitudinal air supply are almost consistent. In addition, the temperature at the safe height is below the critical temperature (68 °C), except for the region near the fire source (−10 m~10 m), and it decreases continuously with the increase in the distance from the fire source. This decrease is mainly attributed to the reduction of radiation from flame. This is mainly because a large amount of high-temperature smoke is accumulated in the region of the fire source and the radiation from the flame, resulting in its high temperature. During the process of smoke movement (including spread and deposition), the high-temperature smoke is continuously mixed with the ambient air and rubbed against the tunnel wall, causing heat exchange and heat loss, hence the temperature of smoke is gradually decreased.

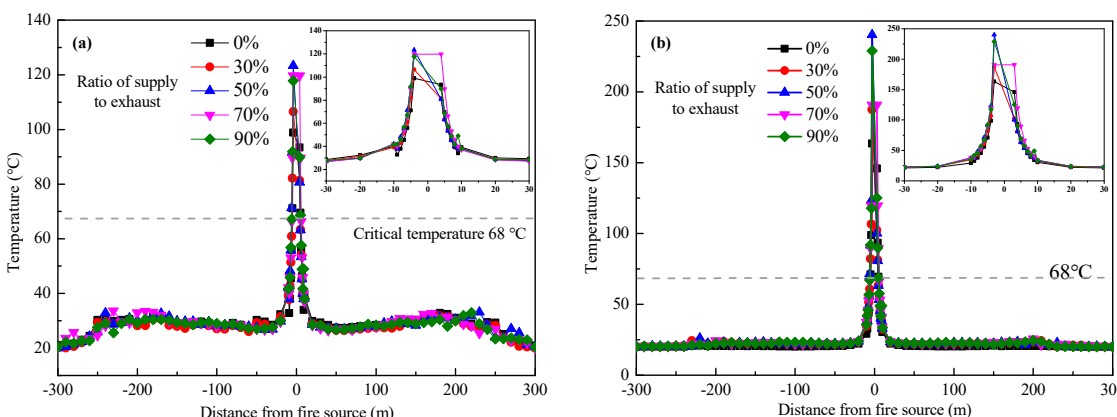

**Figure 5.** Temperature at safe height under different air supply volume: (**a**) sidewall exhaust; (**b**) top exhaust.

However, the influence of the smoke exhaust pattern on the temperature distribution is obvious, mainly including the effects on the fire source region (−10 m~10 m) and the ends of the tunnel. First, the temperature near the fire source for the top exhaust pattern reaches 250 °C, which is much higher than that for the sidewall exhaust pattern (120 °C). This difference is attributed to the accumulation of smoke at the exhaust vents. The smoke spreads upward due to the thermal buoyancy and moves to the top exhaust vents because of the mechanical smoke exhaust, which results in the accumulation of smoke in the tunnel ceiling and a higher temperature in the top exhaust pattern. In addition, for the non-fire region (>10 m), the safe height temperature for the sidewall smoke exhaust (35 °C) is slightly higher than that for the top smoke exhaust (25 °C), where it will not cause harm to escaping people. From the variation of smoke layer height, there is little smoke at the safe height for the top exhaust pattern, and thus the smoke temperature is close to the ambient

temperature. The smoke temperature is raised for sidewall exhaust pattern due to the heat radiation from the lower smoke layer (close to safe height 2 m).

### 3.4. Smoke Exhaust Efficiency

Smoke exhaust efficiency is a main parameter for evaluating the smoke exhaust effect of tunnel fires [34], which is regarded as the ratio of the total smoke exhaust volume to the total smoke generation volume. Since it is difficult to directly obtain the exhaust volume of each exhaust vent in the FDS calculation, the calculation of the exhaust efficiency can be replaced by the mass flux of $CO_2$. Thus, the exhaust efficiency can be calculated as:

$$\eta = \frac{m_e}{m_p} \times 100\% = \sum \eta_i = \frac{\sum m_{ei}}{m_p} \times 100\% \tag{7}$$

where $\eta$ is the exhaust efficiency, %; $m_e$ is the total exhaust rate of $CO_2$ from all exhaust vents, kg/s; $m_{ei}$ is the mass flux of $CO_2$ from $i$-th exhaust vent, kg/s; $m_p$ is the total generation rate of $CO_2$ from fire source, kg/s.

The calculation of the generation of $CO_2$ is complex because of the air entertainment. In the FDS calculation, the fuel can be assumed to be in complete combustion with only a product of $CO_2$, and complete combustion of diesel is simplified as:

$$C_{12}H_{23} + 17.634O_2 \rightarrow 11.868CO_2 + 11.5H_2O + 0.032CO + 0.1Soot \tag{8}$$

The generation rate of $CO_2$ is about 3.32 kg/s, which is calculated from the reaction equation and heat of combustion of 47 MJ/kg [13].

Figure 6 shows the calculated smoke exhaust efficiency with different longitudinal air supply ratios. It is clearly found that the exhaust efficiency for the top exhaust pattern is about 70%, which is significantly higher than that for the sidewall exhaust pattern of 55%, which means the top exhaust pattern has a better smoke exhaust effect. In addition, the longitudinal air supply ratio has a different impact on the smoke exhaust efficiency for different exhaust patterns. For the top exhaust pattern, the exhaust efficiency is improved slowly as longitudinal air supply ratio increases, especially when the longitudinal air supply ratio increases from 30% to 90%, the smoke exhaust efficiency only increases by 2.5 percentage points. Thus, an excessively large amount of air supply will not significantly improve the exhaust efficiency of the top exhaust pattern, and it is reasonable to set the longitudinal air supply at 30%, taking into account the economic cost. In the case of the sidewall exhaust pattern, the longitudinal air supply does not increase linearly with the smoke exhaust efficiency. After the air supply ratio is increased to 70%, the exhaust efficiency will decrease instead. Obviously, air supply of 70% is optimal for the sidewall exhaust pattern.

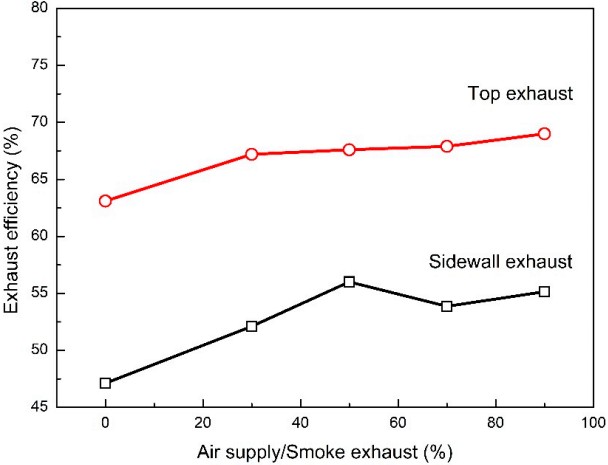

**Figure 6.** Smoke exhaust efficiency under different longitudinal air supply.

## 4. Conclusions

In this paper, a full-scale ultra-wide tunnel model was constructed based on the FDS, with different smoke exhaust patterns (sidewall and top exhaust patterns) and longitudinal air supply ratios (0, 30%, 50%, 70%, and 90%), to explore the smoke control effect for ultra-wide tunnel fires. The fire parameters were analyzed, such as the longitudinal spread distance of smoke, the smoke layer height and the temperature at safe height. In addition, the smoke exhaust efficiency was calculated based on the mass flux of $CO_2$, and the smoke control effect with different smoke exhaust patterns and air supply volume were compared. The main conclusions are summarized as follows:

(1) As a result of the increase in the longitudinal air supply ratio, the smoke spread distance is shortened. The smoke spread distance for the top exhaust pattern is generally shorter than that for the sidewall exhaust pattern, except for the air supply ratio of 70%;

(2) The height of the smoke layer is higher than the safe height of 2 m for the top exhaust pattern, but for the sidewall exhaust pattern, the height is lower than 2 m when the longitudinal air supply volume is less than 50% of the smoke exhaust volume.

(3) The smoke exhaust pattern has a great impact on the temperature near the fire source, which reaches 250 °C for the top exhaust pattern and 120 °C for the sidewall exhaust pattern. In addition, the longitudinal air supply volume has no influence on the temperature, and its distribution is almost consistent.

(4) The exhaust efficiency of the top exhaust pattern (~70%) is significantly higher than that of the sidewall exhaust pattern (~55%). The best air supply ratios are 30% for the top exhaust pattern and 50% for the sidewall exhaust pattern, respectively, taking the economic cost into account.

**Author Contributions:** Conceptualization, Y.L. and K.T.; methodology, Y.L. and C.M.; software, Y.L.; validation, C.M.; formal analysis, Y.L. and F.H.; data curation, Y.L.; writing—original draft preparation, Y.L. and C.M.; writing—original draft preparation, Y.L. and F.H.; writing—review and editing, Y.L. and K.T.; funding acquisition, Y.L. All authors have read and agreed to the published version of the manuscript.

**Funding:** Independent Project Foundation of Naval University of Engineering.

**Data Availability Statement:** Not applicable.

**Conflicts of Interest:** The authors declare no potential conflict of interest with respect to the research, authorship, and/or publication of this article.

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
