# Peer review of "A Simulation Study on the Smoke Control Effect with Different Smoke Exhaust Patterns and Longitudinal Air Supply for Ultra-Wide Tunnels"

_fire, doi:10.3390/fire5030072_

Round 1

Reviewer 1 Report

A full-scale ultra-wide immersed tube tunnel model was constructed to explore the impacts of the smoke exhaust patterns and air supply volume on the effect of smoke control. Sidewall and top smoke exhaust patterns were selected, with different ratios of air supply to smoke exhaust volume. The novelty of this paper is relatively low, and the conducted numerical mode is not properly to simulate a tunnel fire. Some comments are given,

  1. Many previous studies have been conducted and the novelty of this study should be further emphasized.
  2. The author should highlight the major difficulties and challenges in this scenario, and the original achievements to solve them, in abstract and introduction, especially concerting the different from the previous works.
  3. In section 2.1, the author stated that “At the same time, one air supply vent was arranged at both ends of the tunnel to reduce the pressure difference and improve the exhaust efficiency”, this setup is not properly to simulate a tunnel fire. In this condition, it is more of an enclosure fire, the two ends of tunnel should be opened to the environment.
  4. The numerical methods should be verified by comparing with the experimental data and the existing theoretical works.
  5. Most of the major conclusions are easy to think about and lack adequate analysis.

Reviewer 2 Report

Very nice paper but 2 major points to be solved (see detailled comments on enclosed document):

1) the authors missed a set of relevant references in their litterature review.

2) the boundary conditions of the model are not well described.

(See detailled comments on enclosed document.)

Reconsider after major revision

Reviewer 3 Report

Comments:

The authors have investigated the ‘A simulation study on the smoke control effect with different smoke exhaust patterns and longitudinal air supply for ultra-wide tunnel’. Even though the subject matter is interesting; however, this article has not been written systematically and consequently, it  suffers from significant shortcomings and some of them are listed below:

  1. Usually, an abstract of a reader-friendly, scientific article is the ‘self-dependent and concise summary’ of the whole investigation; this is a MANDATORY criterion for writing an abstract. Unfortunately, the current version of the abstract does not satisfy the criterion, as stated above. So, a reader-friendly abstract MUST be written in such a way so that the potential readers, regardless their discipline or expertise, can easily find the following information systematically and easily for example ‘the definition of the problem of investigation’, ‘the precise description of applied methodologies’, and ‘the key findings of the investigation, which are unique and universally valid under the wide range of pertinent conditions. This is one of the major shortcomings of this article.

Besides, some unnecessary sentences/phrases are included in the abstract, which may be appropriate elsewhere, for instance, in the introduction part.   Hence it is recommended to revise the abstract as suggested above.       

  1. The next important and most vital part of an article is the ‘introduction’. It is generally treated as the heart of an article. The introduction part usually guides the flow of the construction of the rest of the article’s parts. So, a question naturally arises on how to construct the introduction of a reader-friendly scientific article? To address the above question, the authors are strongly suggested here to survey the existing literature on the subject matter of the article extensively to reveal the ‘research gap or originality’ within the existing literature. Once the ‘research gap’ has been identified, then the rest of the article MUST be devoted to filling up the ‘research gap’ as identified. Unfortunately, the ‘introduction’ of this article is not written as highlighted above. In other words, the authors have completely failed to reveal the ‘research gap’ in this investigation. Without revealing the ‘research gap’ systematically, any research work has no scientific value! The ‘research gap’ MUST be revealed systematically and it should have both physical and scientific reasons for writing any research gap, which is again an obligatory event! Moreover, a sudden jump in reviewing the literature has been notified in several places which is not expected, and due to the existence of these facts, the current version of the introduction required to be modified. This is a mandatory event for this article.

  1. Since this work is a numerical work, hence it is indispensable to include a section for ‘numerical modelling section’ where governing equations and corresponding boundary conditions MUST be written systematically along with the adopting ‘numerical algorithm and discretizing schemes and so on’ in such as way so that our potential readers can go through this article easily without any difficulties. This is a mandatory event for this article.

  1. The grid-dependency evidence MUST be presented in this article, which will assist the reliability of these numerical results. In addition, no validation is found in this article, which is necessary for any kind of numerical study. These two points are common practice in writing a good, reader-friendly scientific article. Hence, it is strongly recommended to authors to address and reflect them in the revised version.   

  1. Figure 2 (a, b) is not clear, and hence it is required to be modified with the aim of better understanding the Figs and as well as the corresponding text. The same situation prevails for figs 4 and 5.  

  1. Finally, the conclusion part MUST also be precise and straightforward as an abstract so that the potential readers can easily understand events as mentioned above (1) along with major findings of the article. Unfortunately, the ‘conclusion’ part is not written as expected. Besides, the ‘conclusion’ must have consistency with the abstract; this is a common practice of writing a reader-friendly scientific article. Hence, a revision is necessary as suggested above.

  1. This article has other minor problems which MUST be detected by the authors and hence try to address them accordingly. Hence, reviewers would like to see the overall construction of this article with the aim of filling ‘research gap’ as mentioned earlier. Hence, a major revision is necessary.

  1. Anyway, please wait for the comments from the editor.  

Round 2

Reviewer 1 Report

The authors have replied my comments properly, and I suggest the approvement of this paper.

Author Response

Thanks!

Reviewer 2 Report

The main issues raised (lack in litterature review in particular) in the first review are solved.

Author Response

Thanks!

Reviewer 3 Report

1

3   

    Fresh Recommendation:

F   Figure 2 (a, b) is not clear, and hence it is required to be modified with the aim of better understanding the Figs  as well as corresponding to the text.  The same situation prevails for figs 4 and 5.  So, it is a mandatory requirement for getting acceptance letter.  

Author Response

Thanks! We have made the figures as soon as possiable. Please see the attachment.

This manuscript is a resubmission of an earlier submission. The following is a list of the peer review reports and author responses from that submission.